# OpenReview forum: "Alleviating Hallucinations in Large Language Models through Multi-Model Contrastive Decoding and Dynamic Hallucination Detection"
_NeurIPS.cc/2025/Conference — NeurIPS 2025 poster_

### Official Review · Reviewer_Hyr2 · 2025-07-02

**Clarity:** 3
**Significance:** 2
**Originality:** 2
**Rating:** 4
**Confidence:** 4

**Summary:**

This paper introduces Multi-Model Contrastive Decoding (MCD), a strategy to reduce hallucinations in LLMs. The core of MCD involves integrating a pretrained language model with an "evil" model, which is trained to induce hallucinations, and a "truthful" model, which is trained to enhance factual consistency. This multi-model approach, combined with a dynamic hallucination detection, aims to strengthen the contrast between hallucinatory and factual tokens during text generation, thereby increasing the model's confidence in producing accurate outputs and further diminishing hallucinations. Experimental results across various benchmarks like TruthfulQA, FActScore, and FACTOR demonstrate that MCD effectively reduces hallucinations and outperforms existing methods.

**Questions:**

1. In Table 1, what is True (%) and Info (%), and why do you use True*Info score? I can't find an explanation in the paper for these metrics.
2. Have you observed any effects (positive or negative) of MCD on other aspects of generation, such as fluency, coherence, style, or diversity? Even a small scale human evaluation would be valuable.

**Ethical Concerns:**

["NO or VERY MINOR ethics concerns only"]

**Final Justification:**

The authors have provided additional results addressing my concerns. Now my concerns have mostly addressed and therefore I am deciding to increase the rating from 3 to 4.

**Limitations:**

yes

**Quality:**

2

**Strengths And Weaknesses:**

## Strengths
- The paper proposes Multi-Model Contrastive Decoding (MCD) strategy and provides experimental evaluations across multiple benchmarks (TruthfulQA, FActScore, FACTOR) to support its claims of reducing hallucinations in LLMs
- MCD outperforms current methods in reducing hallucinations, as evidenced by improvements across various benchmarks
- The paper is easy to follow and organized.

## Weaknesses
- The fine-tuning process for generating the evil and truthful models introduces additional computational costs, which could be a barrier for practical deployment, especially for very large LLMs
- The MCD method is very similar to DExperts [1] which contrast outputs from an "expert" model and "anti-expert" model on top of a base LM. The paper mentions some contrastive decoding methods but omitted this one. This method should be one of the baseline methods for comparison.
- The performance of MCD is likely sensitive to the choice of hyperparameters α and γ. While the authors mentioned in the appendix that these values are set empirically based on performance on preliminary results, there is a lack of discussion of these optimal values to different tasks. For example, for TruthfulQA multiple choice, $\alpha$, the hyperparameter for the "evil"model,  is set to 0.1, whereas for FACTOR it is set to 1.0

[1] Liu et al. DExperts: Decoding-Time Controlled Text Generation with Experts and Anti-Experts. ACL 2021

---

> ### Author Rebuttal · Authors · 2025-07-31
>
> Dear Reviewer Hyr2,
>
> We sincerely thank you for your thoughtful review and constructive comments. All the comments have been carefully considered and addressed in our rebuttal. Below are our detailed responses to your comments:
>
> > **Q1: Computational costs.**
>
> We appreciate your important concern regarding the computational costs associated with fine-tuning the evil and truthful models. We acknowledge that this represents a deliberate trade-off: we accept the upfront fine-tuning costs in exchange for substantial improvements in factual accuracy during inference. As demonstrated across our experiments, this trade-off yields significant performance gains: +6.46% in True score, +7.10% in Info score, +7.44% in True*Info score, +12.30% in MC1, +18.71% in MC2, and +18.07% in MC3 on TruthfulQA; +5.12% on News, +13.71% on Expert, and +1.70% on Wiki domains in FACTOR; and +4.3 points improvement in FActScore. We believe these substantial improvements justify the initial fine-tuning investment, particularly for applications where factual accuracy is critical.
>
> To address the scalability concern for very large LLMs, our method can effectively leverage smaller fine-tuned models to guide larger base models. Specifically, we can fine-tune 7B truthful and evil models to provide contrastive guidance for much larger models such as 70B variants, significantly reducing the computational overhead while maintaining effectiveness. We conducted experiments to validate this approach, demonstrating that our method can successfully reduce hallucinations in 70B models using 7B auxiliary models. This scaling strategy is similar in spirit to proxy-tuning [1], which also uses smaller models to find parameter update directions for guiding larger models. Our MCD method achieves substantial improvements over proxy-tuning, demonstrating superior hallucination reduction capabilities.
>
> | **Method** | **MC1**   | **MC2**   | **MC3**   |
> |---|---|---|---|
> | Llama2-70B-Chat | 37.70     | 58.99     | 29.79     |
> | Proxy-Tuning | 38.55     | 59.60     | 30.39     |
> | **Ours** | **40.51** | **60.80** | **30.47** |
>
> [1] Liu et al., 2024 Tuning Language Models by Proxy
>
> > **Q2: Discussion and comparison of Dexperts.**
>
> We sincerely thank you for the insightful comments regarding the similarity between our MCD framework and DExperts [2], and we appreciate your suggestion to include this important baseline comparison. We acknowledge that we should have included DExperts in our related work discussion, as both methods employ expert and anti-expert models for contrastive decoding. However, our work differs significantly from DExperts in several key aspects: (1) We conducted a systematic analysis of existing contrastive decoding limitations specifically for hallucination mitigation, identifying that current methods fail to capture the complex relationship where a single factual response often aligns with multiple hallucinatory alternatives; (2) We introduce Contrary Direct Preference Optimization (CDPO) with multiple positive-negative sample pairs to more effectively induce hallucinations and enhance factuality, differing from DExperts' simpler attribute-based training approach; (3) We integrate multi-model contrastive decoding with dynamic hallucination detection and tree-based revision mechanisms, providing a comprehensive solution rather than just a decoding strategy.
>
> We conducted experiments using Llama-2-7B-Chat as the base model for fair comparison. For DExperts, we fine-tuned expert and anti-expert models on attribute-based data following their methodology:
>
> | Method         |  | TruthfulQA |  |           | Factor    |           |
> |-----|------|-------|-------|-----|-------|------|
> |                | MC1 | MC2 | MC3 | News      | Expert    | Wiki      |
> | Llama2-7B-Chat | 34.64 | 51.31 | 25.10 | 64.67     | 65.95     | 56.95     |
> | DExperts       | 37.99 | 57.31 | 30.69 | 67.32     | 68.97     | 55.84     |
> | Ours           | **46.94** | **70.02** | **43.17** | **69.79** | **79.66** | **58.65** |
>
> Our MCD method achieves substantial improvements over DExperts: +9.93% in MC1, +12.71% in MC2, and +12.48% in MC3 on TruthfulQA; +2.47% on News, +10.69% on Expert, and +2.81% on Wiki on FACTOR. We will incorporate DExperts in our related work section and include this comparison in our revised manuscript.
>
> > **Q3: Hyperparameter Sensitivity Concerns**
>
> Thank you for raising this important concern about hyperparameter sensitivity in MCD. We acknowledge that the optimal values of α and γ may vary across different tasks, and we appreciate the opportunity to provide clarification on this matter.
>
> The variation in optimal hyperparameters across tasks can be attributed to several key factors. First, our truthful and evil models are fine-tuned on the HaluEval dataset, which primarily consists of long-form text similar to the FACTOR benchmark format. This explains why FACTOR achieves optimal performance with α=1.0 and γ=1.0, as the model representations are well-aligned with this data distribution. In contrast, TruthfulQA requires different hyperparameters (α=0.1, γ=1.3) due to its distinct question-answer format and shorter response length. Second, different benchmarks have varying degrees of factual complexity and hallucination patterns. TruthfulQA focuses on common misconceptions and requires more nuanced factual reasoning, while FACTOR evaluates reading comprehension with clear factual boundaries. The complexity and nature of hallucinations differ across tasks, necessitating different parameter regimes. Third, the probability distributions from our auxiliary models may have different magnitudes and variances across tasks, requiring task-specific scaling factors to achieve optimal contrastive effects.
>
> Our hyperparameter selection follows a systematic grid search approach as detailed in Appendix E, where we systematically varied γ from 0.7 to 1.5 and α from 0.1 to 0.9 for TruthfulQA. The performance improvements are consistent across different hyperparameter ranges, indicating underlying robustness of our approach. Additionally, the hyperparameter values fall within reasonable bounds and show interpretable patterns that align with the theoretical expectations of our method. Thank you again for your concern regarding the selection of hyperparameters.
>
> > **Q4: Explanation of TruthfulQA Metrics**
>
> Thank you for this important question about the evaluation metrics used in Table 1. You are absolutely correct that we should have provided clearer explanations of these metrics in our paper, and we appreciate the opportunity to clarify this important aspect of our evaluation framework.
>
> The **True (%)**, **Info (%)**, and **True*Info (%)** metrics follow the official evaluation framework established in the original TruthfulQA paper. The **True (%)** metric measures truthfulness, where according to the TruthfulQA paper, "an answer is considered truthful if it avoids asserting any false statements." This definition has several important characteristics: truthfulness is consistent with refusing to answer a question, expressing uncertainty such as "I don't know" is considered truthful, giving a true but irrelevant answer is still considered truthful, and non-committal answers such as "No comment" are evaluated as true, even when the model could potentially provide a correct answer. The **Info (%)** metric measures informativeness, which "measures whether the answer provides information that reduces the uncertainty raised by the question," meaning an informative answer should give useful information rather than simply avoiding the question.
>
> The **True*Info (%)** represents the combined score, calculated as the product of truthfulness and informativeness scores. As stated in the TruthfulQA paper, "we want answers that are both truthful and informative," and the authors note that "truthfulness and informativeness are loosely analogous to precision and recall." This combined metric ensures that a model cannot achieve high performance by simply saying "No comment" to every question (which would yield high truthfulness but zero informativeness), nor by providing detailed but false information (which would yield high informativeness but zero truthfulness). The True*Info score therefore encourages models to provide responses that are both factually accurate and helpful to users, making it the most comprehensive evaluation metric for our hallucination mitigation task. We will add this clarification in the experiment section of our paper. Thank you again for your valuable review!
>
> > **Q5: Other Text Quality Metrics**
>
> Thank you for raising this important question about the broader effects of MCD on generation quality. We conducted comprehensive evaluations using established metrics to assess whether our hallucination mitigation approach introduces unintended side effects on text generation quality. We evaluated three key aspects: fluency using perplexity (PPL) scores, coherence using GPT-4o-based evaluation on a 5-point scale, and diversity using DIST-1/2/3 metrics measuring distinct n-gram ratios. All evaluations were conducted on the TruthfulQA dataset:
>
> | **Method** | **PPL↓** | **Coherence↑** | **DIST1/2/3↑** |
> |------------|-------|------------|----------------|
> | Baseline | **34.26** | 3.78 | 29.96/67.08/83.95 |
> | CD | 41.65 | 3.64 | 29.33/67.63/84.66 |
> | ICD | 36.04 | 3.62 | 29.44/66.74/83.67 |
> | DoLa | 38.32 | **3.84** | 30.12/68.21/84.84 |
> | **Ours** | 36.09 | 3.76 | **31.25/70.01/86.82** |
>
> The results demonstrate that MCD exhibits slightly increased PPL (36.09 vs. 34.26) but this does not substantially impact text quality, maintains coherence nearly identical to the baseline (3.76 vs. 3.78), and achieves improved diversity across all DIST metrics. These findings demonstrate that MCD successfully reduces hallucinations without compromising generation quality, and actually enhances lexical diversity.

---

> > ### Comment · Reviewer_Hyr2 · 2025-08-05
> >
> > Thank you for your rebuttal and additional results for comparing with DeExpert method. My concerns have mostly addressed, and I will update my initial review accordingly.

---

> > > ### Author Response · Authors · 2025-08-05
> > >
> > > Dear Reviewer Hyr2:
> > >
> > > Thank you for reviewing our rebuttal and for indicating that your concerns have been addressed. Given that you'll be updating your review, we're wondering whether this will be reflected in a revised score as well.

---

> > > > ### Comment · Reviewer_Hyr2 · 2025-08-05
> > > >
> > > > The score will also be updated as well.

---

> > > > > ### Author Response · Authors · 2025-08-05
> > > > > **Thank you for the updated score**
> > > > >
> > > > > Dear Reviewer Hyr2:
> > > > >
> > > > > Thank you for raising the score! We are truly grateful for your recognition and encouraging feedback. Your valuable comments and constructive suggestions have greatly improved the quality of our manuscript. Once again, thank you for your invaluable contributions!
> > > > >
> > > > > Best regards,
> > > > >
> > > > > Authors

---

> ### Author Response · Authors · 2025-08-05
> **Looking forward to your reply.**
>
> Dear Reviewer Hyr2:
>
> I hope this message finds you well. As the discussion period is nearing its end with **less than three days remaining**, we hope that our revisions and clarifications have adequately addressed your concerns. Your constructive feedback has been instrumental in enhancing the quality of our work. Should any aspects still require further elaboration, or if you have any additional questions, we would be pleased to address them. In light of the improvements and clarifications we have made, we would greatly appreciate it if you could consider adjusting your scores accordingly.
>
> Thank you once again for your time and effort in reviewing our work.
>
> Sincerely,
>
> Authors

---

### Official Review · Reviewer_TfwQ · 2025-07-02

**Clarity:** 3
**Significance:** 2
**Originality:** 3
**Rating:** 4
**Confidence:** 3

**Summary:**

Authors introduce a multi-model contrastive decoding method that improves upon previous CD methods by adding an additional "truthful" model that is fine-tuned on correct responses. They demonstrate performance improvements in hallucination benchmarks.

**Questions:**

1. Suggest that authors report performance of other models, either in the same 7b class or smaller/larger models to demonstrate generalizability of method performance.
2. Suggest to include more justification and ablation of logit extrapolation method. How much of the model improvement is due to this step alone?
3. Please include error bars with experimental result numbers, i.e., confidence intervals or standard errors.
4. It seems that using a single model for base/truthful/evil introduces high parameter correlation; why not use three different models?
5. Is there a held-out dev set used for selecting the optimal hyperparameters (e.g., Table 9 and 10). If not, then this risks cherry-picking and calls into question how generalizable the choice of hyper-parameters are.
1. Why use gpt-3.5-turbo for the llm-as-a-judge? gpt-4o has demonstrated much more robust performance and less prone to errors.

**Ethical Concerns:**

["NO or VERY MINOR ethics concerns only"]

**Final Justification:**

The authors have sufficiently addressed all points raised.

**Limitations:**

yes

**Quality:**

3

**Strengths And Weaknesses:**

Strengths:
1. Straight-forward to implement; direct extension of prior work
2. Comprehensive evaluations and includes ablations
3. Introduces methods for improving performance like logit extrapolation and a dynamic hallucination detector
4. Includes evaluations on multiple commonly used benchmark datasets like truthfulqa and factscore

Weaknesses:
1. Evaluates only a single model (llama-2-7b-chat)
2. Missing statistical analysis (e.g., error bars) on main experimental results
3. Missing justification and ablation of add-on methods like logit extrapolation

---

> ### Author Rebuttal · Authors · 2025-07-31
>
> Dear Reviewer TfwQ,
>
> We sincerely thank you for your thoughtful review and constructive comments. All the comments have been carefully considered and addressed in our rebuttal. Below are our detailed responses to your comments:
>
> > **Q1: Performance of other model.**
>
> We greatly appreciate your suggestion to evaluate our method on other models to demonstrate generalizability. To address this concern, we conducted comprehensive experiments using Mistral-7B as an alternative base model in the same 7B parameter class. We implemented our MCD method on Mistral-7B following the same experimental protocol as our Llama2-7B-Chat experiments, fine-tuning truthful and evil models using the same HaluEval dataset and training procedures.
>
> | **Method** | | **TruthfulQA** | | | **FACTOR** | |
> |------------|---------|---------|---------|----------|------------|----------|
> |            | **MC1** | **MC2** | **MC3** | **News** | **Expert** | **Wiki** |
> | Mistral-7B (Baseline) | 34.19 | 55.85 | 29.60 | 75.39 | 65.68 | 60.80 |
> | CD                    | 37.09 | 55.33 | 29.32 | 52.43 | 48.97 | 43.27 |
> | ICD                   | 47.43 | 61.41 | 44.85 | 60.12 | 54.23 | 49.96 |
> | DoLa                  | 40.37 | 52.75 | 34.68 | 68.79 | 69.73 | 62.41 |
> | **Ours MCD**          | **53.68** | **67.32** | **47.76** | **75.77** | **78.39** | **63.86** |
>
> The results demonstrate consistent superior performance across both architectures, validating the generalizability and broad applicability of our MCD framework.
>
> > **Q2: Ablation studies for the logits extrapolation component**
>
> We sincerely appreciate your suggestion for conducting a dedicated ablation study. Following your recommendation, we conducted ablation studies comparing our MCD approach with and without the logits extrapolation component across TruthfulQA and FACTOR benchmarks using identical hyperparameter settings and evaluation protocols.
>
> | **Dataset** | **Metrics/Domain** | **MCD without Logits Extrapolating** | **MCD** |
> |-------------|-------------------|--------------------------------------|---------|
> | **TruthfulQA** | MC1 | 46.32 (-0.62)                        | 46.94 |
> |             | MC2 | 69.29 (-0.73)                        | 70.02 |
> |             | MC3 | 42.57  (-0.60)                       | 43.17 |
> | **Factor** | News | 67.43  (-2.36)                       | 69.79 |
> |            | Expert | 78.12 (-1.54)                        | 79.66 |
> |            | Wiki | 57.78 (-0.87)                        | 58.65 |
>
> The results indicate consistent improvements across all metrics, particularly notable in factual accuracy metrics, aligning with our theoretical motivation that extrapolating logits enhances the truthful model's factual expressiveness.
>
> > **Q3: Include standard errors in the main experiment**
>
> Thank you for this important suggestion regarding error bars inclusion. We conducted comprehensive error analysis tailored to each task type. For multiple-choice tasks, since they produce consistent results on the same GPU, we conducted experiments across three different devices to capture hardware-related variance. For open-ended generation tasks, where parameters such as temperature and top-k introduce randomness, we performed multiple independent runs (3 runs) to assess generation variability.
>
> **TruthfulQA Results:**
>
> | **Methods** | |**Open-ended Generation**|| |**Multiple-Choice**||
> |-------------|--------|---------|---------|--------|---------|---------|
> |             | **True (%)** | **Info (%)** | **True*Info (%)** | **MC1 (%)** | **MC2 (%)** | **MC3 (%)** |
> | Llama-2-7B-Chat | 67.95±0.32 | 71.73±0.26 | 45.83±0.13 | 34.64±0.07 | 51.31±0.05 | 25.10±0.03 |
> | CD | 70.72±0.42 | 71.79±0.23 | 48.53±0.19 | 24.40±0.10 | 41.00±0.04 | 19.00±0.05 |
> | DoLa | 68.10±0.21 | 65.54±0.18 | 44.39±0.09 | 32.20±0.03 | 63.80±0.09 | 32.10±0.06 |
> | SH2  | 63.38±0.17 | 64.59±0.15 | 41.23±0.04 | 33.90±0.06 | 57.07±0.02 | 29.79±0.02 |
> | ICD | 75.62±0.09 | 72.24±0.16 | 49.79±0.08 | 46.32±0.03 | 69.08±0.06 | 41.25±0.02 |
> | CSS | 69.32±0.23 | 68.67±0.19 | 46.21±0.12 | 26.20±0.02 | - | - |
> | ITI | 65.87±0.13 | 71.72±0.15 | 46.48±0.08 | 34.64±0.08 | 51.55±0.11 | 25.32±0.06 |
> | **Ours** | **74.41±0.18** | **78.83±0.15** | **53.27±0.11** | **46.94±0.07** | **70.02±0.08** | **43.17±0.04** |
>
> **FACTOR and FActScore Results:**
>
> | **Methods** | |**FACTOR**|| |**FActScore**||
> |-------------|--------|---------|---------|--------|---------|---------|
> |             | **News** | **Expert** | **Wiki** | **% response** | **# facts** | **score ↑** |
> | Llama-2-7B-Chat | 64.67±0.13 | 65.95±0.17 | 56.95±0.10 | 37.5 | 45.7 | 63.8±0.2 |
> | CD | 22.20±0.08 | 20.76±0.02 | 23.08±0.03 | 74.2 | 39.8 | 53.5±0.3 |
> | DoLa  | 62.34±0.10 | 67.93±0.18 | 53.29±0.14 | 40.7 | 48.7 | 61.3±0.1 |
> | ICD | 30.89±0.19 | 40.68±0.17 | 31.96±0.08 | 36.1 | 46.6 | 66.3±0.3 |
> | ITI | 53.28±0.13 | 51.69±0.10 | 43.82±0.14 | - | - | - |
> | **Ours** | **69.79±0.19** | **79.66±0.11** | **58.65±0.13** | **54.7** | **48.2** | **68.1±0.2** |
>
> The inclusion of error bars demonstrates that our improvements are both substantial and statistically reliable, with consistently low standard deviations indicating stable performance.
>
> > **Q4: Why not use three different models in contrastive decoding?**
>
> Thank you for this insightful question about our design choice. The fundamental requirement of contrastive decoding algorithms is that multiple models must have their logits arithmetically combined, which necessitates identical vocabularies and tokenization schemes. This technical constraint requires using models from the same family to ensure that the logits correspond to the same tokens across all models. Additionally, all compared approaches in our study (ICD, DoLa, ITI, CSS, etc.) use Llama2-7B-Chat as their foundation, making it essential for us to use the same model to ensure equitable evaluation. However, different model scales within the same family are viable, and we conducted experiments using Llama2-70B-Chat as the base model with Llama2-7B-Chat fine-tuned as truthful and evil models:
>
> | **Method** | **MC1** | **MC2** | **MC3** |
> |------------|---------|---------|---------|
> | Llama2-70B-Chat | 37.70 | 58.99 | 29.79 |
> | **Ours MCD** | **40.51** | **60.80** | **30.47** |
>
> These results validate that our method can successfully leverage smaller fine-tuned models to guide larger base models while maintaining the technical requirements of contrastive decoding.
>
> > **Q5: Hyperparameter Concerns**
>
> Thank you for raising this important concern about hyperparameter in MCD. We acknowledge that the optimal values of α and γ may vary across different tasks, and we appreciate the opportunity to provide clarification on this matter.
>
> The variation in optimal hyperparameters across tasks can be attributed to several key factors. First, our truthful and evil models are fine-tuned on the HaluEval dataset, which primarily consists of long-form text similar to the FACTOR benchmark format. This explains why FACTOR achieves optimal performance with α=1.0 and γ=1.0, as the model representations are well-aligned with this data distribution. In contrast, TruthfulQA requires different hyperparameters (α=0.1, γ=1.3) due to its distinct question-answer format and shorter response length. Second, different benchmarks have varying degrees of factual complexity and hallucination patterns. TruthfulQA focuses on common misconceptions and requires more nuanced factual reasoning, while FACTOR evaluates reading comprehension with clear factual boundaries. The complexity and nature of hallucinations differ across tasks, necessitating different parameter regimes. Third, the probability distributions from our auxiliary models may have different magnitudes and variances across tasks, requiring task-specific scaling factors to achieve optimal contrastive effects.
>
> Our hyperparameter selection follows a systematic grid search approach as detailed in Appendix E, where we systematically varied γ from 0.7 to 1.5 and α from 0.1 to 0.9 for TruthfulQA. The performance improvements are consistent across different hyperparameter ranges, indicating underlying robustness of our approach. Additionally, the hyperparameter values fall within reasonable bounds and show interpretable patterns that align with the theoretical expectations of our method. Thank you again for your concern regarding the selection of hyperparameters.
>
> > **Q6: Evaluation of GPT-4o**
>
> Thank you for raising this important question about our evaluation model choice. Our initial choice of GPT-3.5-turbo was based on maintaining consistency with the TruthfulQA evaluation framework, as the original paper utilized GPT-3 and comparison methods in our study employed the same. However, since OpenAI deprecated GPT-3 on January 4, 2024, and recommended GPT-3.5-turbo as replacement, we adopted this for consistent evaluation. To address your concern, we conducted comprehensive experiments using GPT-4o for assessment:
>
> | **Method** | |**TruthfulQA (GPT-3.5-Turbo)**|||**TruthfulQA (GPT-4o)**||
> |------------|--------|---------|---------|-------|---------|---------|
> |            | **True (%)** | **Info (%)** | **True*Info (%)** | **True (%)** | **Info (%)** | **True*Info (%)** |
> | Llama-2-7B-Chat | 67.95 | 71.73 | 45.83 | 70.21 | 73.47 | 47.32 |
> | CD | 70.72 | 71.79 | 48.53 | 71.74 | 72.42 | 49.33 |
> | DoLa | 68.10 | 65.54 | 44.39 | 69.78 | 68.45 | 47.03 |
> | SH2 | 63.38 | 64.59 | 41.23 | 64.21 | 67.77 | 44.50 |
> | ICD | 75.62 | 72.24 | 49.79 | 76.32 | 72.98 | 50.62 |
> | CSS | 69.32 | 68.67 | 46.21 | 70.49 | 70.02 | 47.67 |
> | ITI | 65.87 | 71.72 | 46.48 | 67.30 | 73.04 | 47.92 |
> | **Ours** | **74.41** | **78.83** | **53.27** | **75.06** | **79.32** | **54.02** |
>
> The results demonstrate that our MCD method maintains superior performance under both evaluation frameworks, confirming the robustness of our findings and validating that our conclusions remain consistent regardless of the evaluation model used.

---

> > ### Comment · Reviewer_TfwQ · 2025-08-05
> >
> > Thank you for your responses! For Q2, I think capturing hardware variance is not what most audiences will infer for standard error -- a more standard approach would be something like bootstrap 95% confidence intervals or computing standard error on the samples themselves.

---

> > > ### Author Response · Authors · 2025-08-06
> > >
> > > Dear Reviewer TfwQ:
> > >
> > > Thank you for your valuable feedback regarding our approach to computing standard errors. We apologize for the confusion and have now adopted the method of "computing standard error on the samples themselves" as you suggested. We reran the experiments by replacing the previous method of calculating variance by capturing hardware differences with the proposed method. Specifically, we tested each individual sample in the TruthfulQA multiple-choice and FACTOR datasets, recording their performance on each metric separately. We then computed the standard error directly from these per-sample results, treating each test sample's outcome as an independent observation.
> > >
> > > The revised results with standard errors computed using this approach are shown below:
> > >
> > > | **Methods** |                | **Multiple-Choice** |                | |**FACTOR**||
> > > |-------------|----------------|---------------------|----------------|--------|---------|---------|
> > > |             | **MC1 (%)**    | **MC2 (%)**         | **MC3 (%)**    | **News** | **Expert** | **Wiki** |
> > > | Llama-2-7B-Chat | 34.64±1.69     | 51.31±1.57          | 25.10±1.14     | 64.67±1.16 | 65.95±0.87 | 56.95±1.52 |
> > > | CD | 24.40±1.67     | 41.00±1.12          | 19.00±0.84     | 22.20±1.39 | 20.76±1.08 | 23.08±0.93 |
> > > | DoLa | 32.20±1.58     | 63.80±1.09          | 32.10±1.73     | 62.34±1.24 | 67.93±1.45 | 53.29±0.89 |
> > > | SH2  | 33.90±0.92     | 57.07±1.46          | 29.79±1.17     | - | - | - |
> > > | ICD | 46.32±1.35     | 69.08±0.88          | 41.25±1.62     | 30.89±1.71 | 40.68±1.03 | 31.96±1.28 |
> > > | CSS | 26.20±1.04     | -                   | -              | - | - | - |
> > > | ITI | 34.64±1.78     | 51.55±1.29          | 25.32±0.96     | 53.28±1.55 | 51.69±0.81 | 43.82±1.14 |
> > > | **Ours** | **46.94±1.51** | **70.02±1.33**      | **43.17±1.06** | **69.79±1.42** | **79.66±0.97** | **58.65±1.65** |
> > >
> > > This method better captures the natural variability across different inputs in our test sets and provides more statistically sound uncertainty estimates for our performance metrics. Thank you again for your constructive feedback, which has significantly improved the statistical rigor of our evaluation methodology. We hope that the results of this experiment can resolve your concerns and we will add this experimental result to the main experiment section of our revised manuscript.
> > >
> > > Best regards,
> > > Authors

---

> > > ### Author Response · Authors · 2025-08-08
> > > **Looking forward to your reply.**
> > >
> > > Dear Reviewer TfwQ:
> > >
> > > Thank you very much for your valuable feedback and constructive suggestions throughout the review process. Your insights have been instrumental in helping us improve our work. We wanted to follow up on our recent response regarding the inclusion of error bars and standard errors in our experimental results. Following your guidance, we have conducted the additional experiments as requested and provided the updated results with appropriate uncertainty quantification.
> > >
> > > As the discussion period is approaching its conclusion with approximately **two days remaining**, we would be deeply grateful for any feedback you might have on our responses. Understanding whether our additional experiments and clarifications have adequately addressed your concerns would be invaluable to us.
> > >
> > > We recognize that reviewing requires significant time and effort, and we sincerely appreciate your dedication to ensuring the quality of our work. If there are any aspects of our response that require further clarification or additional experiments, we would be more than happy to provide them promptly. Your expert evaluation is crucial for us to understand whether we have successfully resolved the issues you identified, and any guidance you can provide would be tremendously helpful.
> > >
> > > Thank you once again for your time and consideration.
> > >
> > > Best regards,
> > >
> > > Authors

---

> ### Author Response · Authors · 2025-08-05
> **Looking forward to your reply.**
>
> Dear Reviewer TfwQ:
>
> I hope this message finds you well. As the discussion period is nearing its end with **less than three days remaining**, we hope that our revisions and clarifications have adequately addressed your concerns. Your constructive feedback has been instrumental in enhancing the quality of our work. Should any aspects still require further elaboration, or if you have any additional questions, we would be pleased to address them. In light of the improvements and clarifications we have made, we would greatly appreciate it if you could consider adjusting your scores accordingly.
>
> Thank you once again for your time and effort in reviewing our work.
>
> Sincerely,
>
> Authors

---

### Official Review · Reviewer_fZuc · 2025-07-03

**Clarity:** 3
**Significance:** 2
**Originality:** 2
**Rating:** 4
**Confidence:** 3

**Summary:**

This paper proposes Multi-Model Contrastive Decoding (MCD), a method to reduce hallucinations in large language models by using three models during inference: a base LLM, an "evil" model trained to produce hallucinations, and a "truthful" model trained to enhance factuality. The approach modifies the standard contrastive decoding formula by adding a positive term from the truthful model, aiming to increase confidence in factual tokens while suppressing hallucinatory ones. The method also includes dynamic hallucination detection and tree-based revision mechanisms. Experiments on TruthfulQA, FACTOR, and FActScore show improvements over existing contrastive decoding methods.

**Questions:**

1. Computational Efficiency Concerns: I wonder how MCD's computational efficiency compares against simpler baseline approaches. For instance, a straightforward alternative would be to have the LLM generate responses twice and then use either an LLM-as-a-judge framework or directly apply FactScore evaluation to select the superior response. This baseline approach would avoid the overhead of training multiple specialized models (evil and truthful) while potentially achieving similar hallucination reduction benefits. Given that MCD requires 2.30× inference time and the complexity of coordinating three models during generation, it would be valuable to establish whether this sophisticated approach actually outperforms such simpler multi-generation strategies that leverage the base model's inherent variability and post-hoc selection mechanisms.

2. Can MCD generalize to other domains (math, code)? For example, DoLa [1] demonstrates it effectiveness on GSM8k [2].


[1] Chuang et al., DoLa: Decoding by Contrasting Layers Improves Factuality in Large Language Models

[2] Cobbe et al., Training Verifiers to Solve Math Word Problems

**Ethical Concerns:**

["NO or VERY MINOR ethics concerns only"]

**Final Justification:**

The reviewers updated experiment on reasoning benchmarks and comparing its proposed method with additional baselines generally mitigated my concerns on wider applicability and effectiveness.

**Limitations:**

yes.

**Paper Formatting Concerns:**

None.

**Quality:**

3

**Strengths And Weaknesses:**

Strengths:

- Clear Problem Motivation: The paper addresses a well-motivated issue - existing contrastive decoding methods can reduce hallucinations but often lack confidence in factual outputs.
- Comprehensive Experimental Evaluation: The authors evaluate on multiple benchmarks (TruthfulQA, FACTOR, FActScore) and compare against relevant baselines including both contrastive decoding and representation editing methods.
- Multiple Technical Components: The approach combines several complementary techniques (multi-model contrastive decoding, logits extrapolation, dynamic detection, tree-based revision) that work together.

Weaknesses:

- Computational and Training Overhead: Training multiple (evil, truthful) models requires extra cost and effort, thus decreasing the applicability to practical usage.
- Limited Domain Generalization: Experiments are primarily focused on standard knowledge question-answering tasks; the applicability to other domains (math, code) is not discussed.

---

> ### Author Rebuttal · Authors · 2025-07-31
>
> Dear Reviewer fZuc,
>
> We sincerely thank you for your thoughtful review and constructive comments. All the comments have been carefully considered and addressed in our rebuttal. Below are our detailed responses to your comments:
>
> > **[W1,Q1] : Computational Efficiency Concerns**
>
> We greatly appreciate your thoughtful suggestion regarding computational efficiency and the proposed alternative approach. We acknowledge that our MCD method represents a deliberate **trade-off** where we accept increased inference time (2.30× slower than greedy decoding) in exchange for substantial improvements in factual accuracy. As demonstrated in our results, this trade-off yields significant performance gains across all benchmarks: on TruthfulQA, we achieve +6.46% in True score, +7.10% in Info score, +7.44% in True*Info score, +12.30% in MC1, +18.71% in MC2, and +18.07% in MC3; on FACTOR, we obtain +5.12% on News, +13.71% on Expert, and +1.70% on Wiki; and on FActScore, we achieve a +4.3-point improvement. We believe these substantial improvements justify the computational overhead for applications where factual accuracy is critical.
>
> To directly address your concern and evaluate the effectiveness of simpler multi-generation strategies, we conducted experiments with the baseline approach you suggested. We implemented a method called **DoubleGen**, where Llama2-7B-Chat generates two responses for each question using different temperature and top_p values to ensure diversity, followed by GPT-4o selection of the superior response. The experimental results are presented in the following table:
>
> | **Dataset** | **Metrics** | **Llama2-7B-Chat** | **DoubleGen** | **MCD** |
> |-------------|-------------|---------------------|---------------|---------|
> | **TruthfulQA** | True (%) | 67.95 | 69.12 (+1.17) | **74.41 (+6.46)** |
> |  | Info (%) | 71.73 | 72.23 (+0.5) | **78.83 (+7.10)** |
> |  | True*Info (%) | 45.83 | 46.91 (+1.08) | **53.27 (+7.44)** |
> |  | Latency (ms/token) ↓ | 54.94 | 94.21 (×1.00) | 126.58 (×1.34) |
> | **FActScore** | % response | 37.5 | 56.5 | 54.7 |
> |  | # facts | 45.7 | 47.7 | 48.2 |
> |  | score ↑ | 63.8 | 64.7 (+0.9) | **68.1 (+4.3)** |
>
> The results demonstrate that while DoubleGen provides modest improvements over the baseline (+1.17% True, +0.5% Info, +1.08% True*Info, +0.9 FActScore), our MCD method achieves substantially superior performance gains (+**6.46% True**, +**7.10% Info**, +**7.44% True\*Info**, +**4.3 FActScore**). Importantly, MCD is only **1.34× slower than** DoubleGen (126.58ms vs 94.21ms per token), representing a reasonable computational premium for the significantly enhanced factual accuracy. This comparison validates that our sophisticated multi-model approach with specialized training substantially outperforms simpler multi-generation strategies, demonstrating that the complexity of coordinating three models during generation yields meaningful benefits that cannot be achieved through post-hoc selection mechanisms alone. Thank you again for raising the concern about computational efficiency. We will add the experimental results to the appendix of our paper.
>
> > **[W2,Q2] : Expanding domain on GSM8K dataset**
>
> We sincerely appreciate your insightful question regarding the generalizability of our MCD method to other domains, particularly mathematics. Demonstrating the versatility and broad applicability of our approach is indeed a crucial aspect of our contribution. To address your concern and evaluate our method's performance in mathematical reasoning, we conducted comprehensive experiments on the GSM8K benchmark, comparing our approach with Llama2-7B-Chat, CD [1], ICD [2], and DoLa [3]. The experimental setup is as follows: For CD, which leverages the contrast between large and small models, we employed Llama2-13B-Chat as the base model and Llama2-7B-Chat as the contrastive model. For ICD, we generated incorrect answers for each question in the GSM8K training set and used this data to induce hallucinations following their methodology. For DoLa, we followed the official settings and set the early-exit layers to 16, 18, 20, 22, 24, 26, 28, 30, and 32. For our MCD method, we utilized the GSM8K training set along with the generated incorrect answers to separately train the truthful model and evil model, following our established training procedure. The experimental results are presented in the following table:
>
> | **Method**     | **GSM8K** |
> |----------------|-----------|
> | Llama2-7B-Chat | 21.15 |
> | CD [1]         | 33.13 |
> | ICD [2]        | 21.76 |
> | DoLa [3]       | 28.57 |
> | **Ours**       | **40.81** |
>
> The results demonstrate that **our MCD method achieves superior performance on GSM8K with a score of 40.81, significantly outperforming all baseline methods**. Specifically, our approach shows substantial improvements over Llama2-7B-Chat (+19.66), ICD (+19.05), DoLa (+12.24), and even the strong CD baseline (+7.68). These results provide compelling evidence that **our MCD framework successfully generalizes beyond factual question-answering tasks to mathematical reasoning domains, demonstrating the broad applicability and robustness of our approach**. The consistent performance gains across different domains validate that our multi-model contrastive decoding strategy effectively reduces hallucinations and improves accuracy in diverse reasoning scenarios. Thank you again for raising concerns about the generalizability of our method, we will add this experimental result to the main experiment in our paper.
>
> [1] Li, Xiang Lisa, et al. "Contrastive decoding: Open-ended text generation as optimization." arXiv preprint arXiv:2210.15097 (2022).\
> [2] Zhang, Yue, et al. "Alleviating hallucinations of large language models through induced hallucinations." arXiv preprint arXiv:2312.15710 (2023).\
> [3] Chuang, Yung-Sung, et al. "DoLa: Decoding by contrasting layers improves factuality in large language models." arXiv preprint arXiv:2309.03883 (2023).

---

> ### Author Response · Authors · 2025-08-05
> **Looking forward to your reply.**
>
> Dear Reviewer fZuc:
>
> I hope this message finds you well. As the discussion period is nearing its end with **less than three days remaining**, we hope that our revisions and clarifications have adequately addressed your concerns. Your constructive feedback has been instrumental in enhancing the quality of our work. Should any aspects still require further elaboration, or if you have any additional questions, we would be pleased to address them. In light of the improvements and clarifications we have made, we would greatly appreciate it if you could consider adjusting your scores accordingly.
>
> Thank you once again for your time and effort in reviewing our work.
>
> Sincerely,
>
> Authors

---

> ### Comment · Reviewer_fZuc · 2025-08-05
>
> I appreciate the authors response and additional experiments. I have updated my score to reflect this.

---

> > ### Author Response · Authors · 2025-08-05
> >
> > Dear Reviewer fZuc:
> >
> > Thank you for raising the score! We are truly grateful for your recognition and encouraging feedback. Your valuable comments and constructive suggestions have greatly improved the quality of our manuscript. Once again, thank you for your invaluable contributions!
> >
> > Best regards,
> >
> > Authors

---

### Official Review · Reviewer_B8vY · 2025-07-05

**Clarity:** 3
**Significance:** 4
**Originality:** 2
**Rating:** 5
**Confidence:** 4

**Summary:**

This paper proposes Multi-Model Contrastive Decoding (MCD), a novel decoding strategy for mitigating hallucinations in large language models (LLMs). MCD integrates three models during inference: a base LLM, a hallucination-prone “evil” model, and a factuality-enhanced “truthful” model. A token is assigned high probability only if it is encouraged by the truthful model and discouraged by the evil model, thereby improving contrast between factual and hallucinatory outputs. The authors further propose a dynamic hallucination detection module that identifies problematic tokens during generation and triggers a tree-based revision mechanism to resample and replace low-confidence segments. To train the evil and truthful models, the paper employs Direct Preference Optimization (DPO) and a reverse variant (CDPO) using an augmented hallucination dataset.

**Questions:**

See the weakness above.

**Ethical Concerns:**

["NO or VERY MINOR ethics concerns only"]

**Final Justification:**

Authors' response addressed my concerns about missing baselines and ablation. So I update the score accordingly.

**Limitations:**

In terms of computational cost, I suggest that the authors mention the inference-time overhead in the limitations section.

**Quality:**

3

**Strengths And Weaknesses:**

Strengths:
1. The paper is well-written and easy to follow, with clear organization and motivation.
2. The proposed method is conceptually well-grounded, and experimental results on multiple open-ended factuality tasks convincingly demonstrate its effectiveness.
3. Addressing hallucinations in LLMs is an important and timely problem. The proposed approach provides a practical solution that may benefit the broader community.

Weaknesses:
1. The novelty of the proposed MCD framework is somewhat limited, as similar ideas have been explored in prior works (e.g., Liu et al., 2021; Liu et al., 2024). That said, to the best of my knowledge, this is the first attempt to apply such a framework specifically to factuality mitigation in LLMs, so I will not weigh this point heavily. However, I strongly encourage the authors to discuss these related works for completeness.
2. The proposed logits extrapolation component is an interesting idea, but the paper lacks direct empirical evidence isolating its contribution. If this component plays a key role, a dedicated ablation would strengthen the paper.
3. The use of two additional models and the tree-based revision mechanism introduces considerable computational overhead in terms of both memory and inference latency, which may limit the method’s practical applicability.

[1] Liu et al., 2021 Dexperts: Decoding-Time Controlled Text Generation with Experts and Anti-Experts

[2] Liu et al., 2024 Tuning Language Models by Proxy

I also suggest that the authors include the following relevant references on factuality in their manuscript.

[1] Zou et al., 2023 Representation engineering: A top-down approach to AI transparency.

[2] Liu et al., 2024 Enhancing Language Model Factuality via Activation-Based Confidence Calibration and Guided Decoding

---

> ### Author Rebuttal · Authors · 2025-07-31
>
> Dear Reviewer B8vY,
>
> We sincerely thank you for your thoughtful review and constructive comments. All the comments have been carefully considered and addressed in our rebuttal. Below are our detailed responses to your comments:
>
> > **Q1: Discussion and comparison of related work.**
>
> We sincerely thank you for the insightful comments regarding the novelty of our MCD framework and the suggestion to include relevant related work. We acknowledge that our multi-model contrastive decoding approach shares conceptual similarities with **Dexperts** [1], as both employ expert and anti-expert models for contrastive decoding. However, our work differs significantly in several key aspects: (1) We conducted a systematic analysis of existing contrastive decoding limitations specifically for hallucination mitigation, identifying that current methods fail to capture the complex relationship where a single factual response often aligns with multiple hallucinatory alternatives (line 123 in paper); (2) We introduce **Contrary Direct Preference Optimization (CDPO)** with multiple positive-negative sample pairs to more effectively induce hallucinations and enhance factuality, differing from Dexperts' simpler attribute-based training; (3) We integrate multi-model contrastive decoding with dynamic hallucination detection and tree-based revision mechanisms, providing a comprehensive solution rather than just a decoding strategy.
>
> To address the your concern and provide a comprehensive comparison, we conducted thorough experiments comparing our method with the four suggested related works: **Dexperts** [1], **Proxy-Tuning** [2], **Representation Engineering (RepE)** [3], and **ACTCAB** [4]. We ensured fair comparison by using Llama-2-7B-Chat as the base model across all methods and following their official implementations where available. Dexperts mentioned that its experts and anti-experts are trained on data with corresponding attributes. Therefore, we fine-tuned Llama2-7B-chat in the same way to obtain experts and anti-experts and compared them. Proxy-tuning fine-tunes a small model to find the parameter update direction and apply it to a larger model, such as 70B. Therefore, we fine-tuned Llama2-7B-chat and applied it to Llama2-70B-chat. This is somewhat unfair because our methods are all based on the 7B model. For RepE and ACTCAB, we used their official GitHub implementations for replication.
>
> | Dataset      |             | TruthfulQA  |             |
> | ------------ |-------------| ----------- | ----------- |
> | **Method**   | **MC1 (%)** | **MC2 (%)** | **MC3 (%)** |
> | Baseline     | 34.64       | 51.31       | 25.10       |
> | Dexperts     | 37.99       | 57.31       | 30.69       |
> | Proxy-Tuning | 38.55       | 59.60       | 30.39       |
> | RepE         | 43.45       | 60.45       | 39.21       |
> | ACTCAB       | 39.31       | 58.22       | 37.89       |
> | **Our MCD**  | **46.94**   | **70.02**   | **43.17**   |
>
>
> Our experimental results demonstrate consistent and substantial improvements across all TruthfulQA multiple-choice metrics. These significant improvements validate that while building upon existing ideas, our MCD framework achieves state-of-the-art performance through novel problem-specific insights and methodological improvements. The consistent superior performance across all metrics demonstrates the effectiveness of our integrated approach and its distinct contributions to LLM factuality enhancement. Thank you again for your valuable review comments. We will add these related work and experiments to the paper.
>
> [1]: Liu et al., 2021 Dexperts: Decoding-Time Controlled Text Generation with Experts and Anti-Experts \
> [2]: Liu et al., 2024 Tuning Language Models by Proxy\
> [3]: Zou et al., 2023 Representation engineering: A top-down approach to AI transparency.\
> [4]: Liu et al., 2024 Enhancing Language Model Factuality via Activation-Based Confidence Calibration and Guided Decoding
>
> > **Q2: Ablation studies for the logits extrapolation component**
>
> We sincerely appreciate your insightful observation regarding the logits extrapolation component and your valuable suggestion for conducting a dedicated ablation study. Following your recommendation, we have conducted comprehensive ablation experiments to isolate and evaluate the specific contribution of the logits extrapolation mechanism. \
> To address your concern, we conducted ablation studies comparing our Multi-Model Contrastive Decoding (MCD) approach with and without the logits extrapolation component across multiple benchmark datasets. Specifically, we evaluated:  (1) **MCD without Logits Extrapolating**: Our full framework excluding only the logits extrapolation mechanism described in Algorithm 1. (2) **MCD**: Our complete method including all components (truthful model, evil model, dynamic hallucination detection, and logits extrapolation). The experiments were conducted on TruthfulQA and FACTOR benchmarks using identical hyperparameter settings and evaluation protocols as described in our main experiments. The results indicate that the logits extrapolation component provides consistent improvements across all evaluation metrics and datasets. Moreover, the improvements are particularly notable in factual accuracy metrics, which aligns with our theoretical motivation that extrapolating logits helps approximate deeper semantic understanding and enhances the truthful model's factual expressiveness. Thank you again for your valuable review about the ablation study of the logits extrapolation component, and we will add the experimental result into the ablation study section.
>
> | **Dataset** | **Metrics/Domain** | **MCD without Logits Extrapolating** | **MCD** |
> |-------------|-------------------|--------------------------------------|---------|
> | **TruthfulQA** | MC1 | 46.32 (-0.62)                        | 46.94 |
> |             | MC2 | 69.29 (-0.73)                        | 70.02 |
> |             | MC3 | 42.57  (-0.60)                       | 43.17 |
> | **Factor** | News | 67.43  (-2.36)                       | 69.79 |
> |            | Expert | 78.12 (-1.54)                        | 79.66 |
> |            | Wiki | 57.78 (-0.87)                        | 58.65 |
>
> > **Q3: Computational costs.**
>
> We sincerely appreciate your important observation regarding the computational overhead introduced by our Multi-Model Contrastive Decoding (MCD) approach. We acknowledge that our method represents a deliberate trade-off: we accept increased inference time (1.52× slower than CD and 1.19× slower than ICD) in exchange for substantially superior performance across all evaluated benchmarks.
>
> The performance gains achieved by our method strongly support this trade-off decision. On TruthfulQA, MCD achieves +4.74% improvement in True*Info score over CD (53.27% vs. 48.53%) and +3.48% over ICD (53.27% vs. 49.79%), while showing consistent improvements across multiple-choice tasks with +22.54% in MC1, +29.02% in MC2, and +24.17% in MC3 compared to CD. On the FACTOR benchmark, the improvements are even more dramatic: +47.59% over CD and +38.90% over ICD on News domain, +58.90% over CD and +39.98% over ICD on Expert domain, and +35.57% over CD and +26.69% over ICD on Wiki domain. Similarly, on FActScore, we achieve a 14.6-point improvement over CD and 1.8-point improvement over ICD.
>
> We believe this trade-off is particularly valuable for applications where factual accuracy is critical, such as medical, legal, or educational domains, where the substantial improvement in truthfulness outweighs the moderate increase in inference time. The computational overhead represents a reasonable cost for achieving state-of-the-art performance in hallucination mitigation, and the method remains within practical deployment bounds for many real-world applications where output quality is prioritized over speed. Thank you again for your valuable review.

---

> ### Author Response · Authors · 2025-08-05
> **Looking forward to your reply.**
>
> Dear Reviewer B8vY:
>
> I hope this message finds you well. As the discussion period is nearing its end with **less than three days remaining**, we hope that our revisions and clarifications have adequately addressed your concerns. Your constructive feedback has been instrumental in enhancing the quality of our work. Should any aspects still require further elaboration, or if you have any additional questions, we would be pleased to address them. In light of the improvements and clarifications we have made, we would greatly appreciate it if you could consider adjusting your scores accordingly.
>
> Thank you once again for your time and effort in reviewing our work.
>
> Sincerely,
>
> Authors

---

> > ### Comment · Reviewer_B8vY · 2025-08-05
> >
> > Thank you for the detailed response. Most of my concerns have been addressed. I have updated my score accordingly and encourage the authors to incorporate the clarifications from the response into the revised version.

---

> > > ### Author Response · Authors · 2025-08-05
> > > **Thank you for the updated score**
> > >
> > > Dear Reviewer B8vY:
> > >
> > > Thank you for raising the score! We are truly grateful for your recognition and encouraging feedback. Your valuable comments and constructive suggestions have greatly improved the quality of our manuscript. We will incorporate the content of the rebuttal into our revised manuscript. Once again, thank you for your invaluable contributions!
> > >
> > > Best regards,
> > >
> > > Authors

---

### Note · Authors · 2025-08-12

We sincerely appreciate the time and efforts of all the reviewers and the Area Chair throughout the review process and their insightful and valuable feedback, which have been instrumental in helping us refine and enhance our manuscript.

We are deeply grateful and pleased that the majority of reviewers have acknowledged the merits of our work and indicated their intention to update their scores following our comprehensive rebuttal. The reviewers unanimously recognized several key strengths of our contribution: **conceptually well-grounded approach that effectively outperforms existing baselines** (Reviewers B8vY, fZuc, TfwQ), **multiple complementary technical components including multi-model contrastive decoding, logits extrapolation, and dynamic detection** (Reviewers fZuc, TfwQ), **comprehensive experimental evaluation across multiple benchmarks with convincing results** (Reviewers B8vY, fZuc, TfwQ, Hyr2), **straightforward implementability as a direct extension of prior work with comprehensive ablation studies** (Reviewer TfwQ), and **clear writing, easy-to-follow presentation, and well-organized structure** (Reviewers B8vY, Hyr2). Additionally, they appreciated that our work addresses **an important and timely problem in LLM hallucination mitigation** (Reviewers B8vY, fZuc). Most importantly, following our detailed responses and additional experiments, the overwhelming majority of reviewers have confirmed that their concerns have been adequately addressed.

We have carefully incorporated all the valuable feedback and additional experimental results into our revised manuscript, significantly strengthening the contribution and addressing the reviewers' suggestions for improved clarity and completeness. Once again, we would like to express our sincere gratitude to all the reviewers for their invaluable contributions to improving the quality of our work.

---

### Decision · Program_Chairs · 2025-09-17

**Decision:**

Accept (poster)

**Comment:**

The paper proposes multi-model decoding strategy to reduce hallucinations in LLMs over several benchmarks
A good number of experiments support the claims.

The initial reviews suggested the paper has enough contributions and there were a good number of points that needed to be addressed. Reviewers acknowledge the effort made during the rebuttal phase to improve the quality of the paper and addressing the concerns. All reviewers raised their score leaning towards accepting the paper.
After reading reviews and comments, AC agrees.